# Rapid Development of a Theory-Based Targeted Intervention and Communication Plan for HPV Vaccine Introduction in Kosovo Using the Behaviour Change Wheel Model [note 1]

**DOI:** 10.3390/vaccines13080848

**Published:** 2025-08-10

**Authors:** Florie Miftari Basholli, Edita Haxhiu, Isme Humolli, Merita Berisha, Siff Malue Nielsen, Sahil Khan Warsi

**Affiliations:** 1Institute of Public Health, 10000 Pristina, Kosovo; florie.miftaribasholli@uni-pr.edu (F.M.B.); merita.berisha@uni-pr.edu (M.B.); 2Medical Faculty, University of Pristina, 10000 Pristina, Kosovo; humollii@who.int; 3World Health Organization Office, 10000 Pristina, Kosovo; 4World Health Organization Regional Office for Europe, DK-2100 Copenhagen Ø, Denmark; nielsensm@who.int

**Keywords:** HPV vaccine, behavioural insights, intervention development, behaviour change wheel, COM-B, Kosovo

## Abstract

**Background**: Human papillomavirus (HPV) infection is the leading cause of cervical cancer, which presents a significant health burden in low- and middle-income settings such as Kosovo^†^, where it is the second leading cause of death among women. HPV vaccines are highly effective and integral to global cervical cancer elimination efforts. In 2024, Kosovo^†^ introduced the HPV vaccine into its immunisation schedule via a school-based program targeting sixth-grade girls. Rapid, theory-based insights supported development of a tailored communication and intervention plan ahead of the introduction. **Methods**: Over a two-week period, qualitative research was conducted with 102 participants, including healthcare professionals, parents, girls in the target age group, school staff, and community influencers. Data collection, analysis, and intervention development were carried out using the Behaviour Change Wheel (BCW) model, underpinned by the Capability, Opportunity, and Motivation for Behaviour change (COM-B) theoretical framework. **Results**: Trust in school-based immunisation and healthcare professionals emerged as key drivers, while a predominance of capability- and physical-opportunity-related barriers across target groups underscored the need for targeted communication and capacity-building efforts for all stakeholders. Using the BCW model, communication and intervention activities were developed for implementation by partners. **Conclusions**: Using rapid insight research grounded in the BCW model enabled the timely identification of behavioural drivers and barriers to HPV vaccine acceptance and supported development of a targeted intervention plan. The findings echoed global research on HPV vaccine introduction, highlighting context-specific needs and enablers and contributing to a successful rollout marked by high uptake within the first six months.

## 1. Introduction

In 2024, Kosovo^†^ introduced the human papillomavirus (HPV) vaccine into its immunisation schedule in February via a school-based program targeting sixth-grade girls. Ahead of the introduction, originally forecast for October 2023, the Institute of Public Health (IPH), with support from the WHO Office in Pristina and WHO Regional Office for Europe (Regional Office), developed a theory-based HPV vaccine introduction intervention and communication plan (HPV-ICP). The HPV-ICP was developed in two stages of conducting rapid context-specific insight research and then using the results to identify appropriate interventions and collaborating with relevant actors to adapt and finalise the HPV-ICP. Given initial constraints, only two weeks were available to complete this process. Insights, research, and intervention development were guided by the Behaviour Change Wheel (BCW) model.

HPV vaccination is recommended for children aged 9–14, and HPV vaccines have been shown to be safe and effective in decreasing the incidence of cervical cancer in a population, especially when administered to this target group, i.e., before potential exposure to sexually transmitted infections [1,2,3,4]. Cervical cancer is the fourth most common cancer among women globally, with approximately 85% of cases occurring in lower- and middle-income contexts (LMICs) like Kosovo^†^, where cervical cancer is the second leading cause of death among women, with an average burden of 120 cases a year and approximately 89 new cases diagnosed annually in 2020–2022 [2,5,6].

Countries, territories, and areas introducing the HPV vaccine can face several challenges to uptake that must be addressed within context [7]. Low confidence in vaccination or vaccine refusal can be affected by multiple factors, including access barriers, awareness of vaccines and vaccination, and perceptions of vaccine safety and effectiveness [8,9,10]. Additionally, research shows that inclusive strategies that are gender-sensitive and community-specific are essential for promoting vaccine uptake, particularly in settings where variation exists across a population with regard to vaccine confidence, access to services, or sociocultural norms [11,12,13]. Prior to the introduction of the HPV vaccine in Kosovo^†^, limited research indicated low to moderate awareness and support for the HPV vaccine—findings consistent with other Western Balkans contexts, with almost no data available on HPV vaccine introduction interventions specific to the region [14,15,16,17]. In this vein, following broader global evidence, developing a theory-based HPV-ICP based on context-specific insights was seen as crucial to ensuring that efforts targeted the correct audiences with appropriate messages and activities to promote vaccination behaviours leading to greater HPV vaccine uptake [18,19]. By describing the process followed in Kosovo^†^ to develop the HPV-ICP, this article contributes to the growing body of research on HPV vaccine attitudes in the region while also adding to the literature on the effective use of behaviour change theory to develop interventions that address HPV vaccine uptake [20,21,22,23].

## 2. Materials and Methods

### 2.1. Study Design

The two-step process of developing the HPV-ICP was designed using the BCW model: a multi-step model linking identified behavioural barriers and drivers with a set of evidence-based public health intervention types. While multiple theoretical frameworks such as the Behavioural and Social Drivers or Theory of Planned Behaviour can allow for understanding public health behaviours like vaccination, the BCW model provides a comprehensive framework linking behavioural insights to intervention types to support the needs of rapid research with time limitations [24,25,26,27]. Developed and tested on the basis of a review of 19 existing behaviour change intervention frameworks, the BCW model is built around the COM-B theoretical framework of public health behaviour that asserts specific public health behaviours are affected by three interrelated factors, each comprising two sub-factors [28,29]. This framework has been adapted for developing interventions to address vaccine uptake, by focussing on specific factors: capability, physical opportunity, social opportunity, and motivation. Capability relates to how individuals’ knowledge or ability affect performance of a specific behaviour; physical opportunity identifies affordances in the physical context that influence the possibility for performing the behaviour, like access, cost, legislation, etc.; social opportunity concerns aspects of social context, like taboos, mores, discourses, or attitudes, that permit or impede performance of the behaviour; and morality assesses individuals’ instinctive or calculated deliberations on whether or not to perform a behaviour [30]. These COM-B factors are further linked to a set of intervention types shown to address barriers under each factor. Table 1 below provides descriptions of the interventions and which COM-B factor they address.

The first step of the process consisted of a cross-sectional qualitative study on barriers and drivers to HPV-vaccine-uptake-related behaviours for selected target groups. A protocol was developed on the basis of a literature review and consultations with relevant stakeholder representatives, including representatives from the IPH and WHO Office in Pristina and Regional Office technical staff. IPH and WHO ethical approvals were obtained for research, which was conducted across urban and peri-urban neighbourhoods in Pristina, and in two rural locations in the south of Kosovo^†^ where a lack of vaccine uptake had recently been observed. With a two-week limitation for conducting insight research, sample sizes were estimated with consideration of the ‘information power’ of such qualitative research, that is, with the aim to capture pertinent social issues, relationships, and dynamics within and across target groups, by attending to the specificity of aims, targeted sampling of participants, instrumentalisation of the theoretical framework, ensuring of discussion quality, and application of analysis strategy [31]. Employing the principle of information power in qualitative research design ensures that relevant and rich information can be obtained to meet study aims for smaller participant sample sizes. This was observed in the consistency of key themes across sites and target groups, with few new findings emerging after initial discussions.

In the second step, using the BCW model, insight research results on COM-B barriers and drivers to HPV-vaccine-related behaviours for each target group were linked to intervention types. On this basis, the research team drafted a set of potential interventions relating to each intervention type and developed a draft HPV-ICP outlining possible targets, contents, materials required, method of delivery, timeframes, and monitoring requirements for each possible intervention. An initial workshop was held with actors across various sectors relating to HPV vaccine introduction to provide feedback on the draft HPV-ICP. Changes to the draft HPV-ICP were then finalised in a second, smaller workshop with key actors and partners responsible for implementing the final HPV-ICP.

### 2.2. Research Participants

A private research company was contracted to support the recruitment of research participants across four target groups. Health workers, including general practitioners (GPs), paediatricians, gynaecologists, vaccination nurses, and vaccination team members, were randomly selected from publicly available lists for each study site. School staff in the study sites, including primary school principals and teachers of target-aged girls, were similarly selected from publicly available rosters of education professionals. Target-aged girls and mothers and fathers of 12-year-old girls were also randomly selected from the research company’s roster of individuals who had agreed to participate in research. Civil society influencers were purposively sampled to include representatives of organisations working on women and girls’ health across different communities in Kosovo^†^, as well as of underrepresented and marginalised ethnic minority communities.

### 2.3. Ethical Considerations

Written consent to participate in research was obtained from all research participants in Albanian prior to inclusion, including written parental assent for target-aged girls participating in research. As the study aimed to gauge current levels of knowledge about vaccination, participants were informed beforehand that they would be involved in discussing childhood vaccination, but were not informed of HPV vaccine introduction until participation in research activities.

This study was conducted in accordance with the Declaration of Helsinki and approved by the Doctors Chamber Ethical Issues Committee (protocol number 128/23, approved 14 July 2023) and the WHO Research Ethics Review Committee (protocol number ERC.0003986, approved 21 July 2023).

### 2.4. Data Collection and Analysis

Data was collected via a process of simultaneous research and analysis [26]. Focus group discussions (FGDs) and in-depth interviews (IDIs) were conducted using guides developed in English and translated into Albanian. To ensure correspondence across all versions, each guide was reviewed line by line by a group of researchers from the IPH, WHO, and the private research company contracted to organise logistics and moderate research activities. FGDs and IDIs explored drivers and barriers across target groups with regard to physical and social contexts affecting access to general health and vaccination services, as well as individual knowledge, beliefs, attitudes, and practices impacting childhood vaccination behaviours.

FGDs and IDIs were audio-recorded and conducted by a trained moderator. Each activity was observed by an IPH and WHO researcher who took notes on the discussions and participated in moderation to ensure discussion quality [31]. The IPH and WHO researchers convened after the activity or at the end of the day to collectively compile and update a Rapid Assessment Procedure (RAP) sheet for each target group. A RAP sheet is a tool to summarise findings while research is ongoing [32]. A single RAP sheet was compiled for observations relating to each target subgroup. Rows in a RAP sheet were organised by topics from the discussion guide, and columns by the subgroups being compared. Audio recordings of each activity were consulted by the researchers if there was a need to verify a finding. Following this process meant that all RAP sheets were completed with the conclusion of the final research activity, allowing for quick analysis of RAP sheets. Over a day, RAP sheets across target groups were compared and deductively coded to identify thematic COM-B barriers and drivers within and across target groups, both those perceived from outside a target group and those self-reported within a target group [33]. Once a list of barriers and drivers was compiled for each target group, the research team identified which drivers and barriers could be feasibly instrumentalised or addressed, respectively, as part of an intervention prior to HPV vaccine introduction.

Using the BCW model, researchers then specified intervention types connected to the COM-B factors related to the barriers and listed potential feasible interventions for each target group. These potential interventions were then grouped into thematic categories and developed into a table, which served as the basis of the HPV-ICP. For each intervention activity, the table includes target populations, materials needed, any informational or material contents required, method of delivery, timeframe, and intervention monitoring methods.

## 3. Results

### 3.1. Qualitative Insight Research

Over eleven days, 13 FGDs and 6 IDIs were conducted with 102 research participants drawn from across Pristina and selected rural areas. Table 2 below provides a breakdown of participants and activities.

FGDs and IDIs examined the factors affecting health workers’, school staff’s, and civil society influencers’ behaviour of recommending or advising on HPV vaccine uptake, target-aged girls’ parents’ behaviour of vaccinating their daughters, and target-aged girls’ behaviour of getting vaccinated. Across all target groups, 89 COM-B drivers and barriers were identified, of which 60 were considered for intervention development ahead of the HPV vaccine introduction. Only barriers and drivers that could be addressed or instrumentalised in the short term ahead of HPV vaccine introduction were selected for intervention development. This included all identified capability-related barriers, about 90% of identified motivation-related barriers, and about half of identified physical and social opportunity barriers. With the exception of school staff, for each target group, the majority of barriers were related to capability- or physical-opportunity-related issues, reflecting mostly knowledge and skill gaps and barriers to accessing information or services. For school staff, the majority of barriers were related to motivation and capability factors, discussed further below.

Across all target groups excluding health workers, capability barriers primarily included gaps in understanding of how vaccines create immunity, of HPV and cervical cancer, and, especially among male participants, of cervical cancer as a relevant problem for their communities. A major driver was a belief in the validity and authority of information provided by the IPH and health workers. Among health workers, one capability barrier was a lack of information on HPV and cervical cancer incidence and mortality in Kosovo^†^, as well as a general knowledge gap on the HPV vaccine among some health workers like nurses, though a related driver was a belief in the authority of IPH and specialist health workers to provide this information. A capability barrier reported by other target groups was that health workers might believe and share incorrect information on the measles, mumps, and rubella (MMR) vaccine and autism risks, heightening vaccine safety concerns. While very few participating parents had concerns on vaccination and autism, they were aware that some parents were worried about this, and related that some health workers might not know how to respond to parents’ questions on delaying the first dose of MMR vaccine at 12 months or have concerns themselves, furthering parents’ misconceptions. This was not related by health workers, but related to another barrier mentioned by all target groups on health worker skill gaps in vaccine communication with patients, in particular with minority ethnic group parents, those lacking confidence in vaccines, or those perceived as religious.

Physical opportunity barriers across target groups, with the exception of health workers and young girls, included receiving incorrect or insufficient vaccine information from health workers (reported by urban parents), as well as not knowing where to find information on vaccines and seeking vaccination information online via mobile phones. While parents, teachers, and community influencers all trusted health-worker-provided information, they did mention seeking information via mobile phones, not from a particular source, but via Google searches. Particularly with regard to ethnic minority groups, participant-reported barriers included limited access to information or health services due to illiteracy, girls not being enrolled in school, lack of transport, or health worker attitudes. An important physical opportunity driver identified was the pre-existing collaboration among schools, health facilities, and civil society organisations on health topics, particularly the discussion of reproductive health with target-aged girls at some schools. Among parents, a social opportunity barrier observed primarily among fathers and male participants was that HPV and cervical cancer were seen as women’s issues that men would not consider of interest and, especially among rural participants, something that might not be appropriate to discuss in public or mixed company.

For all target groups, the primary motivation barrier was unawareness of information on the safety and effectiveness of the HPV vaccine and evidence for the HPV vaccine beyond general online information, especially regarding evidence for future fertility for vaccinated girls. For health workers, this manifested as a concern around how to communicate with parents on these issues. For school staff, there was an additional motivation barrier around not considering vaccination or health discussions with parents as part of their professional role. For all target groups, a significant driver was trust in information and guidance from the IPH and health workers, and a motivation driver for young girls was the desire to make their own health decisions and know why they are getting vaccinated and how it will help them.

### 3.2. Intervention and Communication Plan Development

As the COM-B factors are interconnected, many of the identified barriers and drivers were related within and across target groups. For example, a capability barrier of weak health worker patient communication skills was related to a motivation barrier of not feeling able to effectively engage with hesitant parents on vaccination, which in turn was connected to the parents’ physical opportunity barrier of not being sufficiently informed during vaccination visits. In a general sense, barriers were related to not possessing sufficient or correct information on vaccination, the HPV vaccine, or cervical cancer; not knowing where to obtain this information or not having access to services or individuals that would support confidence in the HPV vaccine; and wanting to receive detailed and thorough information on HPV vaccine safety and effectiveness from authoritative sources. Insight findings for each target group provided information on the details of what information was needed, why it was required, from and for whom it was required, and how it could best be delivered.

Ahead of intervention development, it was agreed not to consider the intervention types of incentivisation, restriction, and coercion, as the latter two had been shown to potentially result in shorter-term, unsustainable changes in behaviour, the first two were not considered contextually feasible, and incentivisation lacked evidence as an effective vaccination intervention [34,35,36,37]. The intervention types suggested to address the identified COM-B barriers were education, training, persuasion, environmental restructuring, and modelling. After potential interventions were listed for each barrier, they were thematically grouped by activities at the system level, overarching interventions to address shared barriers across groups, target-group-specific interventions based on insights, and communication-based activities. Table 3 provides an overview of the suggested interventions, targets, and related target group COM-B drivers/barriers informing the intervention.

Two system-level activities were developed based on motivation drivers for all target groups, illustrating a need for information and support of stakeholders, the health system level, and international partners involved in the introduction. These activities were developing and distributing an overview document on HPV vaccine evidence, safety, and introduction to relevant actors, and holding a roundtable with these actors to adopt a resolution relating to HPV vaccine introduction and cervical cancer elimination. Three of the five overarching activities were developed as environmental restructuring interventions, i.e., changing the physical or social context by (1) creating and training Kosovo^†^-wide-, regional-, and municipal-level expert spokespersons (medical professionals or other socially prominent individuals) who could support introduction or address any crises as needed; (2) creating a platform or mechanism to allow actors involved in the introduction to access all developed materials for any activities; and (3) holding a press conference on the HPV vaccination launch to raise general awareness and emphasise public authorities’ support for the introduction. An educational intervention of holding a scientific conference was developed to ensure support from the medical community, allaying health worker motivation barriers and addressing capability barriers for narrow specialists, and journalist preparation was suggested as an intervention combining education, training, and modelling to equip journalists with correct information, link them with reliable sources for reporting, and demonstrate best practices for reporting on immunisation.

Specific interventions for health workers, schools, and communities were built to incorporate education, to fill knowledge gaps; training, to address skill gaps in advocating for vaccination; persuasion, to encourage HPV vaccine uptake through communication; and modelling, to provide examples of how to advocate or communicate on vaccination and personal examples to inspire HPV vaccine uptake. Insight research indicated multiple trainings would be required for health workers and different levels of health workers would need to be trained separately, with specialist health workers providing key trainings. Beyond detailed information on the HPV vaccine outlined in Section 3.1 above, health workers would require communication skill training on how to effectively and efficiently address parents’ concerns, engage with hesitant parents, and be aware of any biases that might impede engaging with religious or minority parents. A recently developed WHO training was suggested in which examples and role-plays would allow health workers to acquire and practice communication skills. Similar to health workers, school staff required a similar kind of engagement via trusted health professionals, and findings indicated they would also need time and persuasion to be confident not just in the HPV vaccine, but also in their role as trusted information sources for parents and girls whom they could encourage to get vaccinated.

A set of community-based activities that incorporated education, persuasion, and modelling was also included. It was noted that these would need to be adapted at the local level based on contextually relevant actors. For example, while urban parents might be more sceptical, rural communities might be more sensitive in speaking about cervical cancer or HPV, and minority groups might require alternative communication formats via trusted individuals. Key goals would be to reach fathers as well as mothers—a group not always easy to reach on children’s immunisation—through face-to-face community health worker visits, village council meetings with health workers and civil society organisations, especially those working with minority communities that could face specific challenges in accessing health services and whose daughters might not be enrolled in school. Targeted school-based activities were also suggested for target-aged girls to be engaged by health workers, as well as for parents to learn about and discuss any concerns on the HPV vaccine with a trusted health worker at schools, which would allow engaging both fathers and mothers, especially in urban settings. Beyond information dissemination, research insights specified that parents would be persuaded through health workers’, civil society influencers’, or other trusted actors’ confidence in explaining and presenting details on evidence of HPV vaccine safety, effectiveness, and necessity.

In this vein, a set of communication activities was built around environmental restructuring to make currently unavailable information available with content and formats suited to parents and others, persuasion to encourage HPV vaccine uptake through presenting targeted messaging, and modelling through stories of vaccinated women and parents who chose to vaccinate their children against HPV. A key intervention was the creation of a dedicated website for vaccination that would provide key details and evidence for childhood vaccines, including the HPV vaccine, accessible by mobile phones, and referenced across all materials developed. Print, video, and social media materials, as well as the production of TV talk shows with specialist health workers and health officials, were suggested with key messages from findings targeting mothers, fathers, girls, health workers, and the general public. A final suggested environmental restructuring intervention was the conducting of social media monitoring to support timely identification and addressing of mis- or disinformation risks or other crises. Upon conclusion of the second workshop, administrators overseeing the HPV vaccine introduction indicated all suggested HPV-ICP interventions would be incorporated into the broader introduction planning.

## 4. Discussion

Research on employing the BCW model for intervention design has been limited, with studies primarily focussing on using the model to analyse health behaviour and developed interventions [38,39]. While limited, studies on utilising the model for public health intervention design or strengthening, including for addressing HPV or other vaccine uptake, have mostly been conducted in the Global North, with some research also conducted with populations in or from LMICs. This research has reported on processes spanning periods between six months and multiple years [23,40,41,42,43]. The HPV-ICP development process detailed in this article contributes to this literature by describing the use of the BCW model for rapid qualitative research to develop interventions in an LMIC-comparable context, while also addressing a gap in the literature on HPV vaccine uptake in the Southern Balkans.

As this study was conducted in the wake of the COVID-19 pandemic, circumstances required the process of insight research and intervention development to be completed in two weeks. In this context, rapid research methods were used that have been proven effective and rigorous, but also shown to be open to challenges including small sample sizes, limited time for data triangulation, and lower granularity of research data [24,44]. The study did indeed face recruiting challenges with parents and girls, where FGDs were planned for 5–8 participants, each representing a different demographic within the study site, but not all of those recruited were able to participate on the day, and, given time constraints, research could not be replanned. To counteract such expected challenges, guidance on increasing information power of qualitative research had been followed by employing an overall theoretical framework and engaging trained qualitative researchers with local and subject area expertise to participate in each stage of the process, while also involving decision makers and potential intervention implementers in the process where and as feasible [31]. Thus, while insight results were limited in not being representative across Kosovo^†^, they did give an indication of the dynamics and issues affecting HPV vaccine uptake that could be addressed via interventions. Insight research results did reflect global findings on behavioural factors affecting HPV vaccine uptake, specifically how preparation and engagement of health workers, including narrow specialists, and teachers was vital for school interventions, and that beyond addressing knowledge gaps and emphasising actors’ influential role for parents, bolstering effective communication skills and strategies to address parents’ questions and concerns was crucial [45,46,47,48].

Findings across all target groups on safety and effectiveness concerns echoed global research highlighting not just people’s anxieties around the HPV vaccine’s potential effects on fertility, but also their challenges to accessing trusted information, and social taboos around discussing women and girls’ health in public or mixed settings [49,50,51,52]. The finding that all target groups, with the exception of health workers, would need more than just IPH approval of the HPV vaccine to be confident, but would also require detailed information on the vaccine’s safety, including contents, scientific evidence, longitudinal research results, etc., is also in line with research showing perceived lack of information on HPV vaccine as the most frequent concern reported across studies on HPV and other vaccine confidence [53,54]. While other research suggests such concerns on HPV vaccine safety are more connected to low trust in authorities, in this context, authorities enjoy high public trust, evidenced in research but also by almost all study participants indicating trust in IPH vaccine decision making [53,55,56]. Insight research limitations precluded exploring this issue in further depth. It is possible that this need for more detailed scientific information on the HPV vaccine relates to changes in post-pandemic vaccine-information-seeking behaviour or in perceptions of authoritative safety evidence for vaccines perceived as ‘new’ or that it could reflect the means through which authorities’ engagement supported increases in COVID-19 vaccine uptake [57].

While the BCW model facilitated identifying the intervention types, study results suggested that individuals influential at the community level (health workers, community leaders, etc.) would need to be involved in delivering interventions. Other research has shown that while no significant evidence exists on effectiveness of interventions, even specifically for vaccination, tailoring activities to the local context and simple action-oriented interventions, like reminder/recall strategies, are effective [37,58,59]. The need for local-level engagement was particularly salient for the identified need to engage with men on the HPV vaccine. While studies have emphasised the need for vaccinating boys and men and increasing their HPV awareness, to our knowledge, no research exists on effective interventions to engage men as vaccine advocates [60,61,62]. Research suggests gender-specific strategies relevant to social context, such as using male community leaders as messengers, framing vaccination in terms of protecting loved ones, and leveraging peer norms among men, could be promising approaches to engage men more effectively in vaccine advocacy and uptake [11,63,64].

While efforts were undertaken to minimise study limitations and the use of rapid research methods permitted the development of an evidence-based HPV-ICP within the short period of two weeks, time limitations did impact intervention development. Data analysis was focussed primarily on identifying intervention types related to COM-B barriers, with intervention details depending on researcher experience and stakeholder feedback. In the case of the suggested intervention on community engagement, the draft HPV-ICP had to just indicate that the intervention would require local-level development. Had more time been available, additional analysis could have been conducted with insight data using the Theoretical Domains Framework (TDF) to consider specific behaviour change techniques (BCTs) linked to each of the COM-B barriers identified [65,66].

## 5. Conclusions

Rapid research methods were successfully used in tandem with the BCW model for intervention design to develop an intervention and communication plan for HPV vaccine introduction within a limited period of two weeks. A two-step process was used to first identify COM-B barriers and drivers for relevant target groups via qualitative insight research and then identify feasible related intervention types, finalising the plan in conjunction with actors responsible for intervention implementation. Most barriers were interconnected around issues of knowledge and skill gaps, highlighting the need to fully prepare health professionals and school staff ahead of the introduction and ensure that more detailed information on the HPV vaccine and evidence of its safety and effectiveness were available to the public. Findings also emphasised the importance of community-level tailoring of interventions to local contexts, especially to reach particular populations like men or ethnic minority groups, who may face specific or unique barriers to vaccination behaviours. Interventions largely focussed on education as well as environmental restructuring and modelling to ensure access to information and uptake of requisite skills and vaccination behaviours. Study limitations were mitigated by ensuring research team expertise, involvement in each stage of the process, and investment of time over the two weeks available, though time limitations precluded detailed intervention development using specific behavioural change techniques.

## Figures and Tables

**Table 1 vaccines-13-00848-t001:** BCW model intervention types and linked COM-B factors.

	COM-B Factors			
Intervention Types	Capability	Physical Opportunity	Social Opportunity	Motivation
Education: Increase knowledge or understanding	✓	✕	✕	✓
Persuasion: Using communication to induce positive or negative feelings or stimulate action	✕	✕	✕	✓
Incentivisation: Creating expectation of reward	✕	✕	✕	✓
Coercion: Creating expectation of punishment or cost	✕	✕	✕	✓
Training: Imparting skills	✓	✓	✕	✓
Restriction: Using rules that limit engagement in the target behaviour or competing or supporting behaviour	✕	✓	✓	✕
Environmental Restructuring: Changing the physical or social context	✕	✓	✓	✓
Modelling: Providing an example for people to aspire to or imitate	✕	✓	✓	✓

“✓” indicates a intervention type appropriate for the COM-B factor, and “✕” an inappropriate intervention type.

**Table 2 vaccines-13-00848-t002:** Research participants and activities conducted by target group.

			Participants	
Target Groups		Activity	Urban and Peri-Urban	Rural
Health workers	GPs, paediatricians, gynaecologists	2 FGDs	*n* = 10 (7F, 3M)	*n* = 6 (1F, 5M)
	Nurses (vaccination)	2 FGDs	*n* = 10 (10F)	*n* = 8 (2F, 6M)
	Mobile vaccination team members	1 FGD	*n* = 16 (16F)	--
Parents and girls	Mothers of target-aged girls	2 FGDs	*n* = 5 (5F)	*n* = 6 (6F)
	Fathers of target-aged girls	2 FGDs	*n* = 4 (4M)	*n* = 4 (4M)
	12-year-old girls	2 FGDs	*n* = 6 (6F)	*n* = 4 (4F)
Social influencers	School staff (principals)	2 IDIs	*n* = 1 (1M)	*n* = 1 (1M)
	School staff (teachers)	2 FGDs	*n* = 9 (9F)	*n* = 7 (4F, 3M)
	Civil society influencers	4 IDIs	*n* = 3 (2F, 1M)	*n* = 2 (1F, 1M)
	Total	13 FGDs 6 IDIs	*n* = 64 (55F, 9M)	*n* = 38 (18F, 20M)

**Table 3 vaccines-13-00848-t003:** Suggested interventions, targets, and related insight target group COM-B factors.

Suggested Intervention	Intervention Target (*Intervention Type*)	Insight Research Target Group (*COM-B Factor*) Informing Intervention
**System-Level Activities**		
HPV vaccine introduction document	Health system and international partner representatives (*education*)	Health workers (*physical opportunity*, *motivation*) Parents (*motivation*) School Staff (*motivation*) Civil society reps (*motivation*)
Round table and resolution on HPV vaccine introduction and cervical cancer elimination	Health system and international partner representatives (*education*)	
**Overarching Activities**		
Kosovo^†^-wide, regional, and municipal expert spokesperson group (medical professionals, influential actors)	General public (*environmental restructuring*)	Health workers (*physical opportunity*, *motivation*) Parents (*motivation*) School Staff (*motivation*) Civil society reps (*motivation*)
Intersectoral collaboration and exchange platform/mechanism	Actors supporting HPV vaccine introduction (*environmental restructuring*)	Parents (*motivation*) School Staff (*motivation*) Civil society reps (*motivation*)
HPV vaccine scientific conference	Scientific and medical community, civil society, journalists (*education*)	Health workers (*capability*, *physical opportunity*) Civil society reps (*motivation*)
Vaccine launch press conference	Health workers, civil society, public (*environmental restructuring*, *education*)	Health workers (*motivation*) Parents (*motivation*) School Staff (*motivation*) Civil society reps (*motivation*)
Journalist preparation	Journalists (*education*, *training*, *modelling*)	Parents (*capability*, *physical opportunity*, *social opportunity*)
**Research-Target-Group-Specific Activities**		
Tailored community engagement	Parents, girls, minority groups, rural/urban populations, civil society organisations (*education*, *persuasion*, *modelling*)	Parents (*capability*, *physical opportunity*, *social opportunity*, *motivation*) Civil society reps (*capability*, *physical opportunity*, *social opportunity*)
School staff awareness and communication training	School principals, teachers, teacher associations, municipal directorates *(education, training, modelling*)	School staff (*capability*, *physical opportunity*, *social opportunity*, *motivation*)
School-based activities for parents and girls	Parents, girls (*education*, *persuasion*)	Parents (*capability*, *physical opportunity*, *social opportunity*, *motivation*) Girls (*capability*, *motivation*)
Health worker awareness and communication trainings	GPs, nurses, narrow specialists, public health experts (*education*, *training*, *modelling)*	Health workers (*capability*, *physical opportunity*, *motivation*) Parents (*physical opportunity*, *social opportunity*, *motivation*)
**Communication Activities**		
Communication materials (insight-based) Dedicated vaccines website, print materials, video PSAs, TV talk-shows, social media tiles	Public, health workers, girls (*environmental restructuring*, *education)*	Health workers (*capability*, *physical opportunity*) Parents (*capability*, *physical opportunity*, *motivation*) Girls (*capability*, *motivation*) Civil society (*capability*, *physical opportunity, motivation*)
Social media monitoring	Actors supporting HPV vaccine introduction (*environmental restructuring*)	Health workers (*physical opportunity*, *social opportunity*) Parents (*capability social opportunity*) Civil society (*capability*, *social opportunity*)

## Data Availability

Data in the form of RAP sheets and RAP sheet analysis are available on request from S.M.N. until 2028 when they will be destroyed as per conditions of ethical approval.

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
