# Peer review of "Rapid Development of a Theory-Based Targeted Intervention and Communication Plan for HPV Vaccine Introduction in Kosovo Using the Behaviour Change Wheel Model [Author-notes fn1-vaccines-13-00848]"

_vaccines, 2025, doi:10.3390/vaccines13080848_

Round 1

Reviewer 1 Report

Comments and Suggestions for Authors

The research design type is a cross-sectional study. The manuscript described a practice of applying behavior change theory (BCW model) to rapidly develop an intervention program for HPV vaccine rollout, with a clear study design and sound application of the theory, which is an important reference for vaccine rollout in low- and middle-income areas (e.g., Kosovo). The study was completed in a very short period of time (two weeks) for data collection, analysis, and intervention design, demonstrating efficient execution, but some of the methodological details and depth of results could be further optimized. The researchers interview 102 people to understand the barriers to HPV vaccination in the 12-year-old girl population. The study involved four populations including healthcare professionals, parents, target age girls, school staff, and community influencers, ensured that the intervention design took into account both system-level and community-level needs. The targeted intervention plan developed in this study emphasizes background needs and support, which will contribute to the promotion and application of HPV vaccines.

Major shortcomings and suggestions for revision

1.There is no mention of how data saturation will be ensured.please describe how to get the sample size.

2.The text lists "target-aged girls’ parents", "target-aged girls", "health workers", "social influencers", and "school staff". How were "social influencers" defined/recruited?

3.Use “IPH” and “IHP” accurately, especially on lines 198 and 201, and ensure that the correct acronyms are used throughout (IPH - Institute of Public Health? IHP - Incorrectly Headed?)

Comments on the Quality of English Language

English writing needs further improvement.

Reviewer 2 Report

Comments and Suggestions for Authors

The article titled "Rapid development of a theory-based targeted intervention and communication plan for HPV vaccine introduction in Kosovo" describes how Kosovo introduced the HPV vaccine in 2024 using a school-based program for sixth-grade girls. A rapid, two-week qualitative study was conducted with 102 participants, including health workers, parents, students, and community influencers. Using the Behaviour Change Wheel (BCW) and COM-B framework (Capability, Opportunity, Motivation for Behaviour), the study identified barriers and drivers of vaccine uptake. They discussed: High trust in health professionals and school-based vaccination; Barriers such as knowledge gaps, misinformation, cultural taboos, and access issues, especially in rural and minority communities and interventions focused on education, training, communication, and community engagement, tailored to specific target groups.

The resulting HPV vaccine intervention and communication plan was successfully integrated into Kosovo’s broader immunization efforts, achieving high initial uptake. The study highlights the effectiveness of theory-driven, context-specific planning under time constraints.

  1. Introduction

Highlight the need for inclusive, gender-sensitive, and community-specific strategies upfront.

  1. Materials and Methods

Broaden Stakeholder Sampling: Include more fathers, ethnic minorities, and male community leaders in focus groups and interviews. Use purposive sampling to capture underrepresented voices.

Digital and Literacy Accessibility: Incorporate digital literacy assessments to tailor communication materials. Add alternative communication formats (e.g., audio, video, pictorial).

Health Worker Assessment: Introduce a structured assessment of misinformation prevalence among health workers to better target training content.

  1. Results

Engagement of Fathers & Minority Groups: Report specific attitudes and barriers from male participants and ethnic groups.

Misinformation Mapping: Present a breakdown of misconceptions (e.g., about MMR-autism) across stakeholder groups.

Digital Access & Literacy: Detail findings on information-seeking behavior (e.g., reliance on mobile phones vs. healthcare providers).

  1. Discussion

Male Engagement Strategies: Emphasize the untapped role of men in vaccine decision-making and suggest gender-specific outreach.

Behavioral Nudges: Recommend low-cost nudging strategies (e.g., school-based reminders, visual commitment tools).

Monitoring Gaps: Acknowledge limited monitoring plans and propose real-time feedback mechanisms.

  1. Conclusion

Call for Targeted Adaptations: Stress the importance of localized, inclusive, and gender-responsive strategies.

Plan for Continuous Improvement: Recommend ongoing feedback collection and adaptive intervention refinement.
